# Rescuing historical weather observations improves quantification of severe windstorm risks

Ed Hawkins[1], Philip Brohan[2], Samantha N. Burgess[3], Stephen Burt[1], Gilbert P. Compo[4,5], Suzanne L. Gray[6], Ivan D. Haigh[7], Hans Hersbach[3], Kiki Kuijjer[7], Oscar Martínez-Alvarado[1,6], Chesley McColl[4,5], Andrew P. Schurer[8], Laura Slivinski[4,5], Joanne Williams[9]

[1] National Centre for Atmospheric Science, Department of Meteorology, University of Reading, Reading, UK
[2] Met Office Hadley Centre, Exeter, UK
[3] Copernicus Climate Change Service, ECMWF, Reading, UK
[4] Cooperative Institute for Research in Environmental Sciences, University of Colorado at Boulder, Boulder, USA
[5] NOAA Physical Sciences Laboratory, Boulder, USA
[6] Department of Meteorology, University of Reading, Reading, UK
[7] School of Ocean and Earth Science, National Oceanography Centre, University of Southampton, Southampton, UK
[8] School of Geosciences, University of Edinburgh, Edinburgh, UK
[9] National Oceanography Centre, Liverpool, UK

*Correspondence to*: Ed Hawkins (ed.hawkins@ncas.ac.uk)

**Abstract.** Billions of historical climatological observations remain unavailable to science as they exist only on paper, stored in numerous archives around the world. The conversion of these data from paper to digital could transform our understanding of historical climate variations, including extreme weather events. Here we demonstrate how the rescue of such paper observations has improved our understanding of a severe windstorm that occurred in February 1903 and its significant impacts. By assimilating newly rescued atmospheric pressure observations, the storm is now credibly represented in an improved reanalysis of the event. In some locations this storm produced stronger winds than any event during the modern period (1950-2015) and it is in the top-4 storms for strongest winds anywhere over land in England & Wales. As a result, estimates of risk from severe storms, based on modern period data, may need to be revised. Examining the atmospheric structure of the storm suggests that it is a classic Shapiro-Keyser-type cyclone with 'sting jet' precursors and associated extreme winds at locations and times of known significant damage. Comparison with both independent observations and qualitative information, such as photographs and written accounts, provides additional evidence of the credibility of the atmospheric reconstruction, including of sub-daily rainfall variations. Simulations of the storm surge resulting from this storm show a large coastal surge of around 2.5m, comparing favourably with newly rescued tide gauge observations and adding to our confidence in the reconstruction. Combining historical rescued weather observations with modern reanalysis techniques has allowed us to plausibly reconstruct a severe windstorm and associated storm surge from more than 100 years ago, establishing an invaluable end-to-end tool to improve assessments of risks from extreme weather.

## 1 Introducing Storm Ulysses

Extreme wind events are among the costliest natural disasters in Europe. Significant effort is dedicated to understanding the risk of such events, usually using observed storms in the modern era (e.g. Roberts et al. 2014), and synthetic event sets or ensemble seasonal hindcasts designed to sample a wider range of plausible storms (e.g. Sharkey et al. 2020; Walz & Leckebusch 2019). Severe historical storms that occurred before around 1950 are largely unstudied because atmospheric reanalyses usually only cover the modern era and atmospheric reanalyses that do exist for earlier periods may not represent severe storms plausibly due to the sparseness of the observations available to constrain the atmospheric circulation. However, it is likely that some earlier historical windstorms are more extreme and/or follow different tracks from those in the modern era. Expanding the numbers of reconstructed severe historical storms will improve our understanding of the risks from such events today and in the future.

Achieving this goal requires making more historical observations available to be used in reanalyses by (1) improving access to already digitised observations, and (2) extracting additional observations from archival material. Here we demonstrate how the digitisation of weather observations from paper archives has improved the reconstruction of one particular extreme storm, and enabled the creation of a credible reanalysis of the event.

Between 26-27th February 1903 a violent windstorm passed across Ireland and the UK, causing many deaths, several shipwrecks, and considerable damage to infrastructure. For example, the Royal National Lifeboat Institution (RNLI) recorded 10 major rescues of crew from ships in distress, and The Times newspaper reported damage across the country, with considerable numbers of injuries and loss of life. In a special report on the event, Shaw (1903) described locations where damage or casualties occurred, both on land, and at sea. Figure 1 reproduces the summary figure from Shaw (1903), which also indicates the estimated path of the storm.

Figure 2 includes three photographs showing trees uprooted in Dublin (Ireland), damage to a pier in Morecambe, and a train blown over in Cumbria (both in NW England). A written account of the storm experienced in Carlisle (NW England) is also included. The damage in Ireland even inspired a passage in the novel *Ulysses*, written by James Joyce, with the events set the year after the storm in 1904:

*O yes, J.J. O'Molloy said eagerly. Lady Dudley was walking home through the park to see all the trees that were blown down by that cyclone last year and thought she'd buy a view of Dublin.*

To pay homage, this windstorm is called Storm Ulysses (Met Eireann, 2017).

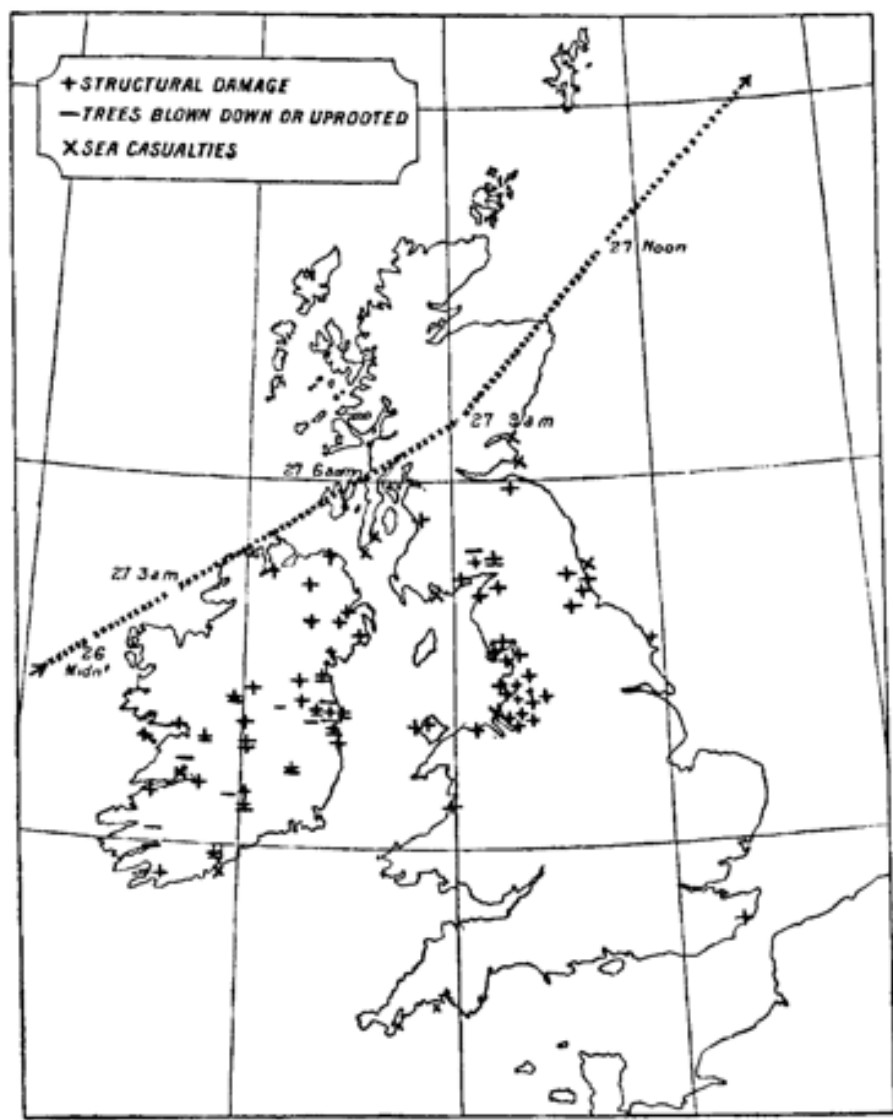

**Figure 1: Post-storm estimate for the track of Storm Ulysses and locations of damage caused. Map taken from Shaw (1903).**

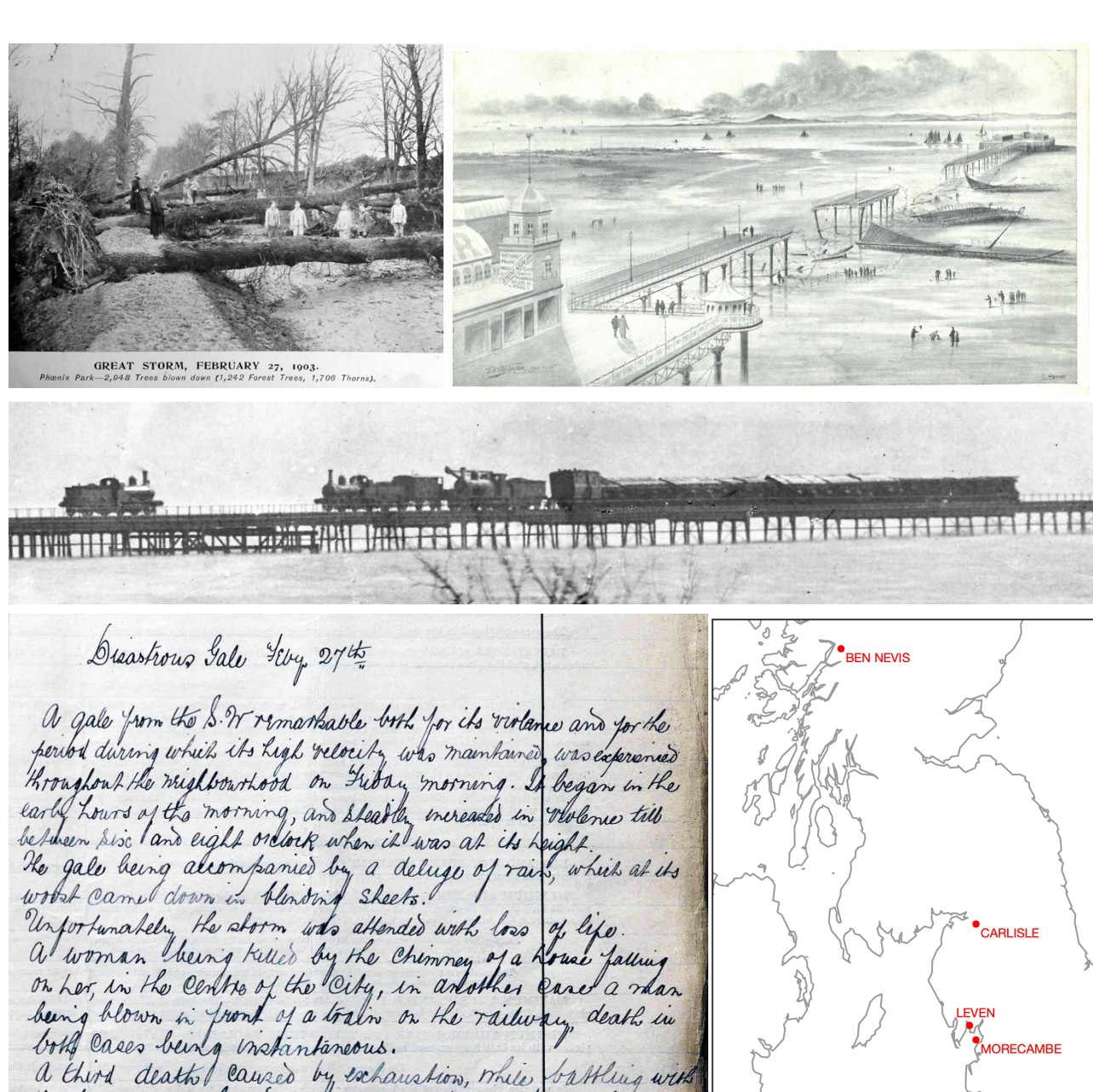

**Figure 2: Visual descriptions of damage from Storm Ulysses. Top: photographs from Dublin (left) and Morecambe (right). Middle: photograph of a train blown over on Leven viaduct. Bottom: written account of the storm in Carlisle & a map of locations in the photos or named in the text. Dublin image supplied by Aida Yared. The Leven photograph was taken by a Mr Alexander, assistant engineer for the Furness railway. The Morecambe image is a scan of a postcard owned by one of the paper authors.**

70

## 2 Reconstructing Storm Ulysses

Modern dynamical reconstructions of historical windstorms rely on reanalyses that assimilate observations of surface pressure that were taken at the time, over both land and ocean, into an atmospheric model, in a similar process to making the initial conditions for a modern weather forecast. In this study we use the NOAA-CIRES-DOE 20th Century Reanalysis version 3 system (20CRv3; Compo et al. 2011; Slivinski et al. 2019a; Slivinski et al. 2021), which has previously produced atmospheric reconstructions for the 1806-2015 period at 0.7° horizontal resolution, generating 3-hourly data, with 80 ensemble members to sample uncertainty in the reconstruction.

We perform novel experiments with this reanalysis system to demonstrate the value of assimilating additional surface pressure observations to better constrain the atmospheric circulation during Storm Ulysses. We evaluate the reanalyses against independent observations and then use the reanalyses to drive a storm surge model and compare against newly rescued tide gauge observations.

Figure 3a shows the synoptic situation according to 20CRv3 (ensemble mean) at 0900 UTC on 27th February 1903, with a low-pressure cyclone situated over the British and Irish Isles. However, the reanalysis is uncertain about some details of the synoptic situation, with regions of >7 hPa ensemble standard deviation over northern UK (Figure 3d). The depth of the low in the ensemble mean (967 hPa) is shallower than an estimate made soon after the event (around 960 hPa; Shaw 1903), but note that the minimum pressure in individual ensemble members is 960 ± 9 hPa (one standard deviation), highlighting that position and timing uncertainty is making the storm appear shallower in the ensemble mean. The red dots in Figure 3d represent the locations of available pressure observations which are assimilated between 0600 UTC and 1200 UTC on this day to produce the reanalysis. These observations are relatively sparse, preventing the reanalysis from being able to accurately identify the location of the low pressure and hence represent the severity of the storm. For example, there were no available observations over England or Wales. This is a common feature of such reanalyses when examining extreme events occurring many decades ago (Brönnimann et al. 2013; Meyer et al. 2013) and currently limits the usefulness of such reconstructions for examining individual severe weather events.

However, since the International Surface Pressure Database (Cram et al. 2015) version 4 (Compo et al. 2019) used within 20CRv3 was assembled, two citizen science projects have rescued additional pressure observations for this period and region which can be used to improve the reanalysis. Thousands of volunteers transcribed millions of meteorological observations from scanned copies of paper records (Hawkins et al. 2019; Craig & Hawkins 2020) and a few additional records have been digitised specifically for a short period around this event. In total, pressure observations from 89 locations have been added (60 over the British & Irish Isles), with most providing two observations per day (see Appendix A for more details).

The 20CRv3 system has been used to repeat the assimilation process for Storm Ulysses, including these new observations. An additional experiment was performed that also included a small improvement to the data assimilation scheme which ensured that the 20CRv3 ensemble was more representative of the uncertainty (see Appendix B for more details). Figures 3b and 3c show the synoptic situation in the improved reanalysis experiments; note an additional isobar highlighting a deeper low pressure which is

more consistent with the estimate made at the time. Across the ensemble members in these two additional experiments, the minimum low pressure depths are 960 ± 5 hPa and 960 ± 3 hPa respectively, highlighting the improved confidence in the position and timing of the storm. The isobars are also closer
together over the British and Irish Isles, meaning that the highest wind speeds over both land and sea have increased by 15-20%. The increased density of available observations (Figure 3e; dark blue dots) has reduced the uncertainty in the reconstruction, and the ensemble spread is further reduced when the assimilation scheme is improved (Figure 3f), becoming more reliable when compared with independent data (see Appendix B).

Figure S1 shows the mean sea level pressure evolution of the storm in 20CRv3 and the two experimental versions of the reanalysis. The experiments with additional observations show minimum pressures around 956 hPa at slightly earlier times than shown in Figure 3.

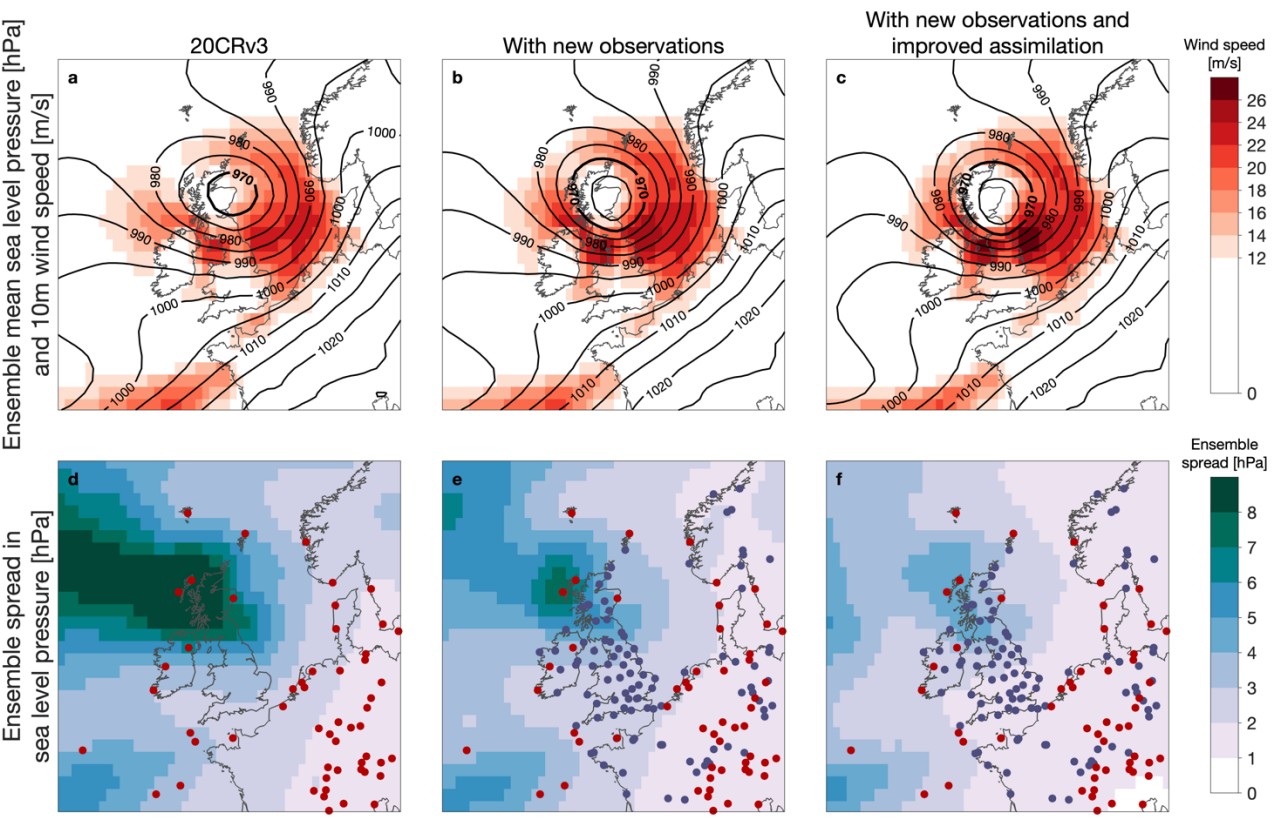

**Figure 3: Reconstructing the atmospheric circulation during Storm Ulysses.** Synoptic situation at 0900 UTC on 27[th] February
1903. Isobars of sea level pressure (hPa; black contours) and wind strength at 10m (m/s; red shading) from the ensemble mean of 80 reanalysis fields are shown from 20CRv3 and the two different experiments (top row; a-c). Standard deviation of the ensemble of sea level pressure reanalysis fields ('ensemble spread') for the same time (hPa; blue shading; bottom row; d-f). Locations with available surface pressure observations in 20CRv3 are shown as red dots and new added observations in the experiments are shown as dark blue dots. Observations from both land stations and ships are shown, but there are very few available ship
observations in this region at this time.

# 3 Reconstructions of wind speeds and the atmospheric circulation

## 3.1 Surface winds

To examine the severity of this storm in more detail we first consider the near-surface wind speeds. Figure 4a-c shows the wind footprints of Storm Ulysses for 20CRv3 and the two experiments; these are maps of the ensemble mean maximum 10m wind speed experienced at each location using instantaneous 3-hourly reanalysis data during the storm. Gust strength data is not available from the reanalysis. The two experiments produce higher wind speeds, particularly in areas of known significant damage such as eastern Ireland and northern England (see Figure 1). Given that damage related to wind is approximately proportional to the cube of the wind speed (Lamb 1991; Klawa & Ulbrich 2003), even a small increase in wind strength can result in significantly increased storm damage.

To quantify the relative strength of these simulated winds it is necessary to compare against other windstorm events in the same reanalysis. We chose to compare with all events during 1950-2015 in 20CRv3 as this represents the period typically available from commonly used reanalyses of the modern period (such as ERA5; Hersbach et al. 2020). If a historical storm is unusual relative to this modern period, then it adds significant information about windstorm risk. Note that 20CRv3 does not yet extend beyond 2015.

Figure 4d-f shows the ranking of the winds experienced during Storm Ulysses compared to all events during the 1950-2015 period. For the original 20CRv3 reanalysis, Storm Ulysses is not particularly unusual, with wind speeds in the top-10 events for some small areas (Figure 4d). However, when the reanalysis is better constrained by additional observations, Storm Ulysses is in the top-5 strongest wind events for larger areas across the UK, Ireland, and the North Sea (Figure 4e). Once the assimilation process is also improved, the reanalysis of Storm Ulysses produces the strongest winds of any event for some locations (Figure 4f), demonstrating the value of having additional observations to constrain the atmospheric circulation to better understand risks.

When looking across the whole of England & Wales, the peak 10m wind speed over land during Storm Ulysses in the improved reanalysis is similar to the 3 most severe storms in the modern era, as represented by 20CRv3. Those storms occurred in 1990 (Burns Day storm), 1997 (Yuma) and 1998 (Fanny), each affecting a slightly different part of the country. Note that this comparison is restricted to the ensemble mean of simulated 10m winds from instantaneous 3-hourly data and does not account for gusts, so may miss some of the most extreme winds from any particular storm (e.g., the 1987 Great Storm). If also including both Ireland and Scotland, the Boxing Day storm of 1998 produced stronger winds than Storm Ulysses in this reanalysis. Regardless of the precise rankings, Storm Ulysses is an extreme windstorm in the context of the modern era, and we can now say that with confidence, even though it occurred over 100 years ago.

This type of historical information is highly relevant to sectors such as insurers, who need to understand the risks of extreme windstorms over the ocean (Buchana & McSharry, 2019) and over the land (Koks & Haer, 2020). Windstorm catalogues (e.g., Roberts et al. 2014) tend to consider the more recent period only, although it is recognised that this may not give a complete picture (Zimmerli & Renggli, 2015).

Incorporating detailed information from significant historical storms such as Storm Ulysses is likely to improve estimates of windstorm risk.

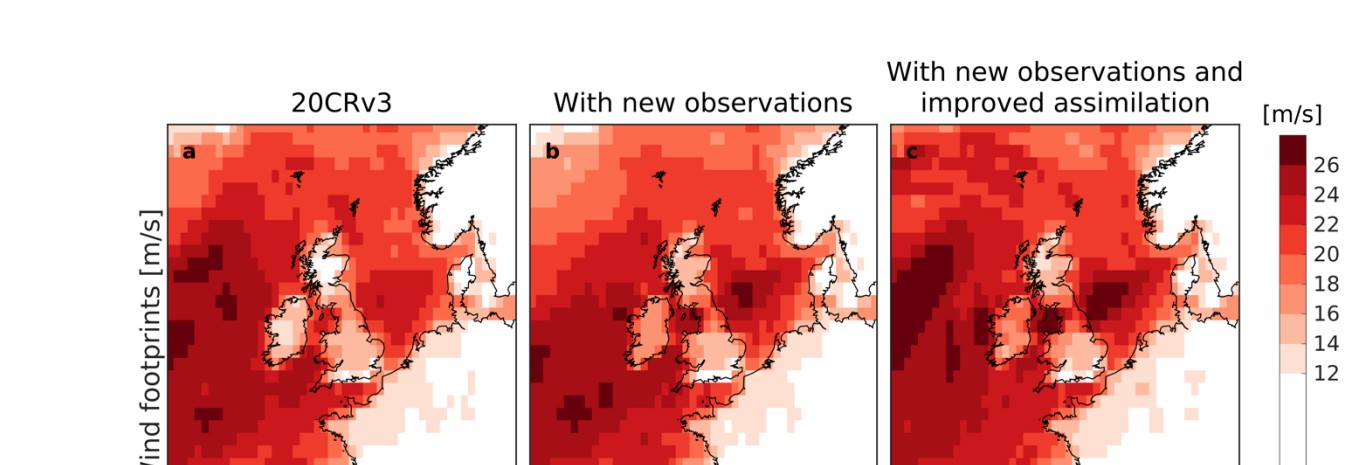

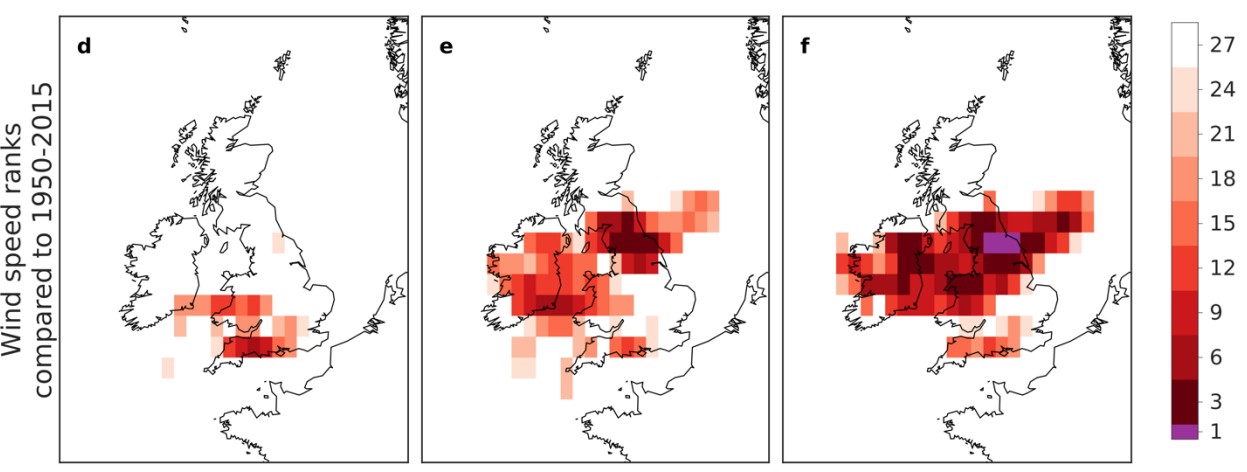

**Figure 4: Comparing wind speeds with other events in the modern era. Wind footprints (top row, m/s; a-c; the maximum 10m wind speed experienced at each location using instantaneous 3-hourly reanalysis data during the storm) and ranking of 10m wind speed compared to all events during 1950-2015 (bottom row; d-f) for Storm Ulysses. The columns show 20CRv3 and the two experiments with the reanalysis system. Purple colours indicate locations where Storm Ulysses would have been the strongest observed had it occurred during 1950-2015.**

## 3.2 Atmospheric conditions and sting jet precursors

Although only surface pressure observations are assimilated, they can substantially constrain the lower part of the atmosphere in the reanalyses, and the three-dimensional structure of the storm provides valuable information. We first examine 850hPa wet-bulb potential temperature ($\theta_w$) during the storm,

with a focus on 0600 UTC on 27th February (Figure 5). This quantity is commonly used to identify warmer and cooler airmasses. High values of relative humidity (RH) at 700hPa are indicated by stippling to highlight the approximate location of the cloud head. The features visible, such as the
hooked cloud head and developing warm seclusion, indicative of frontal fracture, are consistent with a classic Shapiro-Keyser-type cyclone (Shapiro & Keyser 1990). These features are more pronounced in the improved reanalysis experiments and Figures S2 & S3 show their development during the storm.

Figure 5 (bottom row) shows wind speed at 850hPa and highlights two separate regions of higher wind speeds; this level was chosen to avoid contamination from strong orographic signals. In all the
reanalyses there is an extended area of strong winds in the cyclone's warm sector ($\theta_w > 284$ K) but in the experiments, the strongest winds occur in an apparent frontal fracture zone, just to the south of the low pressure centre at the rear of the cold front, and extending rearwards from this as the storm develops (see Figures S4 and S5). Such strong winds found to the cold side of the bent-back front, that lies along the inner edge of the cloud head, are typically attributable to the cold conveyor belt jet. As
this jet extends to the south of the storm, the alignment with the storm's direction of travel yields strong Earth relative winds. These intense wind jets are typical for this type of storm but are not present in the ensemble mean of 20CRv3. However, a small number of individual ensemble members in 20CRv3 do have a coherent wind jet in this region.

There are not usually observations of wind speed at 850hPa anywhere for this historical period, but
there is one existing high-frequency record from the period of Storm Ulysses to which we can compare the reanalyses at this height. Meteorologists living at an atmospheric observatory on the summit of Ben Nevis (1345m above sea level, at 56.8ºN, 5.0ºW) recorded detailed weather observations manually every hour from 1883-1904, including temperature, rainfall, pressure, wind speed and wind direction (Hawkins et al. 2019). The hourly pressure observations from this observatory are included as some of
the new observations added into the reanalysis. This observatory was usually at a height of roughly 850hPa, but during Storm Ulysses the observed pressure fell to 810hPa at 0500 UTC. The summit observers measured force 10-11 winds from 0200-0300 UTC on 27th February which, on the extended wind scale used, is equivalent to around 45 ms$^{-1}$ (Hawkins et al. 2019). The improved reanalysis shows the highest 850hPa wind speeds of 28-38 ms$^{-1}$ (5-95% range) at 0300 UTC on 27th February, whereas
20CRv3 simulates 11-40 ms$^{-1}$ (5-95% range) for the same time. Although this is only a single location, it is encouraging agreement on the timing of peak winds at this elevation. It is difficult to evaluate the amplitude of the wind speeds given that the reanalysis has a coarse resolution relative to the orography in this region, but the improved reanalysis appears more consistent with the available observations.

The wind speed at 850hPa is often used as an estimate of maximum surface gust speed (Hart et al.
2017) so it can also be compared to information about known damage near sea level. It is notable that the train on the Leven viaduct (Figure 2) was blown over at 0530 UTC (Board of Trade, 1903), consistent with the timing of 850hPa winds, and therefore potential surface gusts, of above 40 ms$^{-1}$ in that region in the improved reanalyses (Figure 5). 20CRv3 does not simulate such strong winds at this location.

## Storm Ulysses: February 27th 1903 at 0600 UTC

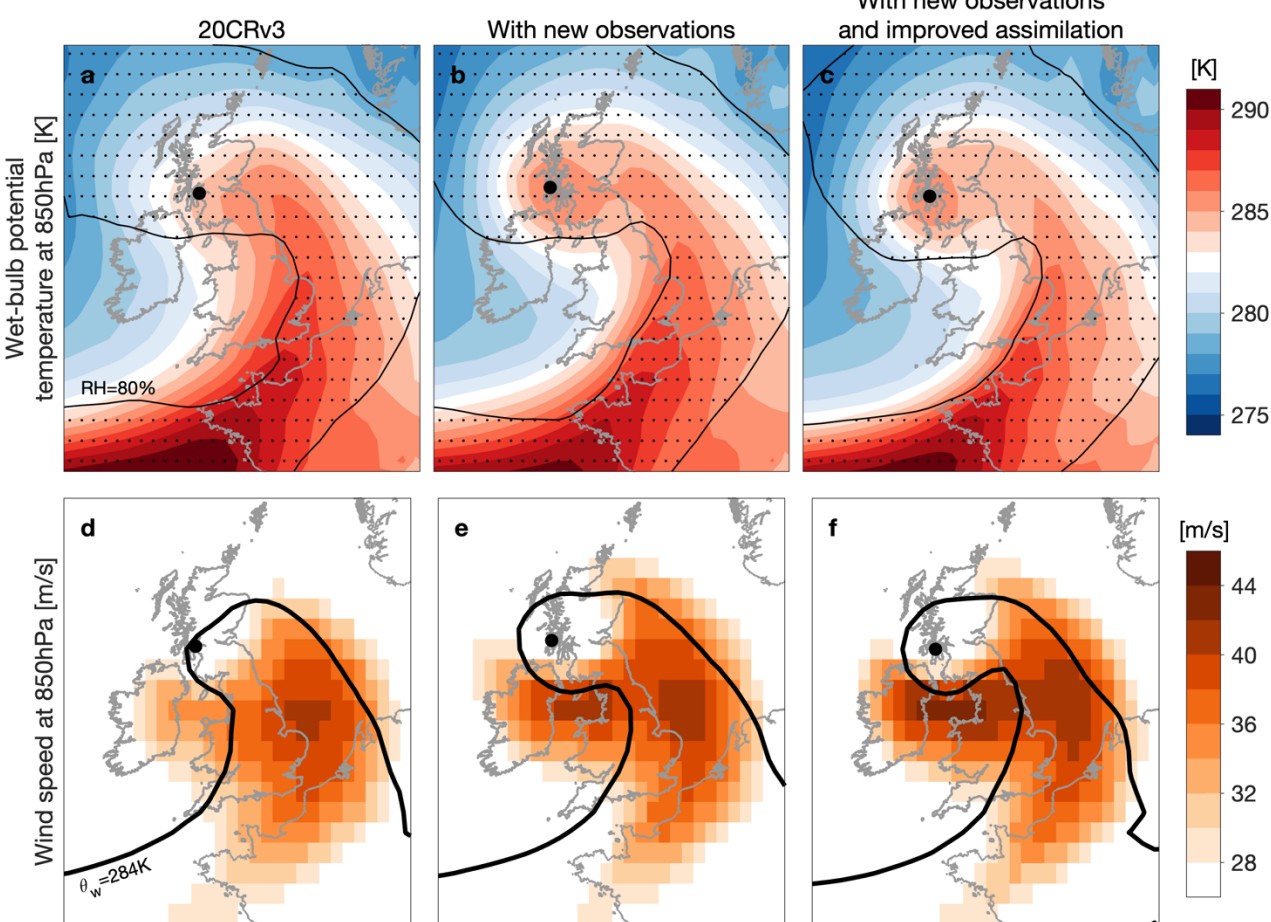

**Figure 5: Ensemble mean of wet-bulb potential temperature at 850hPa (top row; a-c) and wind speed at 850hPa (bottom row; d-f) in the original 20CRv3 and two experiments with the reanalysis system (columns) at 0600 UTC on 27th February 1903. The black filled circle represents the position of the mean sea level pressure minimum. The top panels also include stippling and a contour representing the location of the cloud head using relative humidity (RH) with respect to ice of above 80% at 700hPa. The 284K isotherm is indicated with the thick black contour in the bottom panels.**

It is often the case that the greatest damage from Shapiro-Keyser windstorms comes from meso- and convective- scale phenomena, and this type of cyclone is known to produce sting jets. A sting jet is "a coherent air flow that descends from mid-levels inside the cloud head into the frontal-fracture region of a Shapiro-Keyser cyclone over a period of a few hours leading to a distinct region of near-surface stronger winds" (Clark & Gray 2018; after Browning 2004). These small-scale features cannot be explicitly resolved in the relatively low-resolution model used to generate the available reanalyses, but a metric has been developed for diagnosing precursor conditions suitable for sting jet formation (Martínez-Alvarado et al. 2012). Mesoscale instability release has been shown to occur in storms with intense sting jets (Gray et al. 2011; Volonté et al. 2018) and the precursor metric assesses the presence

in the storm's cloud head of a type of mesoscale convective instability called conditional symmetric instability using a diagnostic called DSCAPE (downdraught slantwise convective available potential energy). The metric was shown to be skilful in identifying storms (from low-resolution model output) in which sting jets developed in corresponding high-resolution simulations capable of resolving mesoscale instability release (Martínez-Alvarado et al. 2012), and it is now applied routinely by the Met Office to provide information relevant to issuing severe wind warnings (Gray et al. 2021).

Figure 6 (top row) shows the track of the storm (defined as the location of minimum interpolated sea level pressure every 3 hours) and the number of ensemble members in which DSCAPE is above 200J/kg (a typical threshold that identifies a sting jet precursor, while also requiring RH > 80% at the level where the DSCAPE threshold is exceeded). Previously a threshold on the number of neighbouring grid points, or grid points within a neighbourhood, in the cloud head with significant DSCAPE has been used as an indicator of the likelihood of a sting jet (e.g., Gray et al. 2021). Instead, we adopt a simpler approach by calculating the fraction of ensemble members with one or more grid points where the DSCAPE threshold is reached to determine the ensemble probability of the presence of mesoscale convective instability. The number of ensemble members with this precursor, and hence the probability of a sting jet, increases as the reanalysis improves. Over half (55%) of the ensemble members in the improved reanalysis show some precursors during the storm at locations in the cloud head to the north-west of the track of the storm. These precursors appear several hours before the strongest winds are observed to the south of the low pressure, as typically seen in such storms. In 20CRv3, only 30% of ensemble members show such a precursor. For the DSCAPE precursor likelihood, there is a clear difference between the experiments that only differ due to the assimilation scheme changes (39% vs 55%). We suggest that this may be because DSCAPE is a threshold-based binary metric meaning that the reduction in ensemble spread has a larger effect.

High DSCAPE values have been previously found to be an indicator for strong surface winds in such datasets (Hart et al. 2017; Clark & Gray, 2018). This can be examined for Storm Ulysses by splitting the reanalysis ensembles into two. The maximum wind speeds at 850hPa in the ensemble members with a sting jet precursor are clearly larger than in those members without a sting jet precursor in each of the reanalyses (Figure 6, bottom two rows). Ensemble mean wind speeds of >45m/s (100mph) are simulated across members of the improved reanalysis with a precursor and occur at locations where known significant damage occurred (Figure 1). The members without a precursor have significantly lower wind speeds and the ensemble mean does not reach 45m/s.

Overall, the improved reanalysis appears to be a credible representation of the storm. ]. Simple comparisons have highlighted the value of both quantitative observations and qualitative information to evaluate the plausibility of the reanalyses. The photographic and written evidence is notable for enabling an evaluation of the reconstructions for aspects of the storm for which detailed instrumental measurements are not available.

# Storm Ulysses: sting-jet precursor metrics

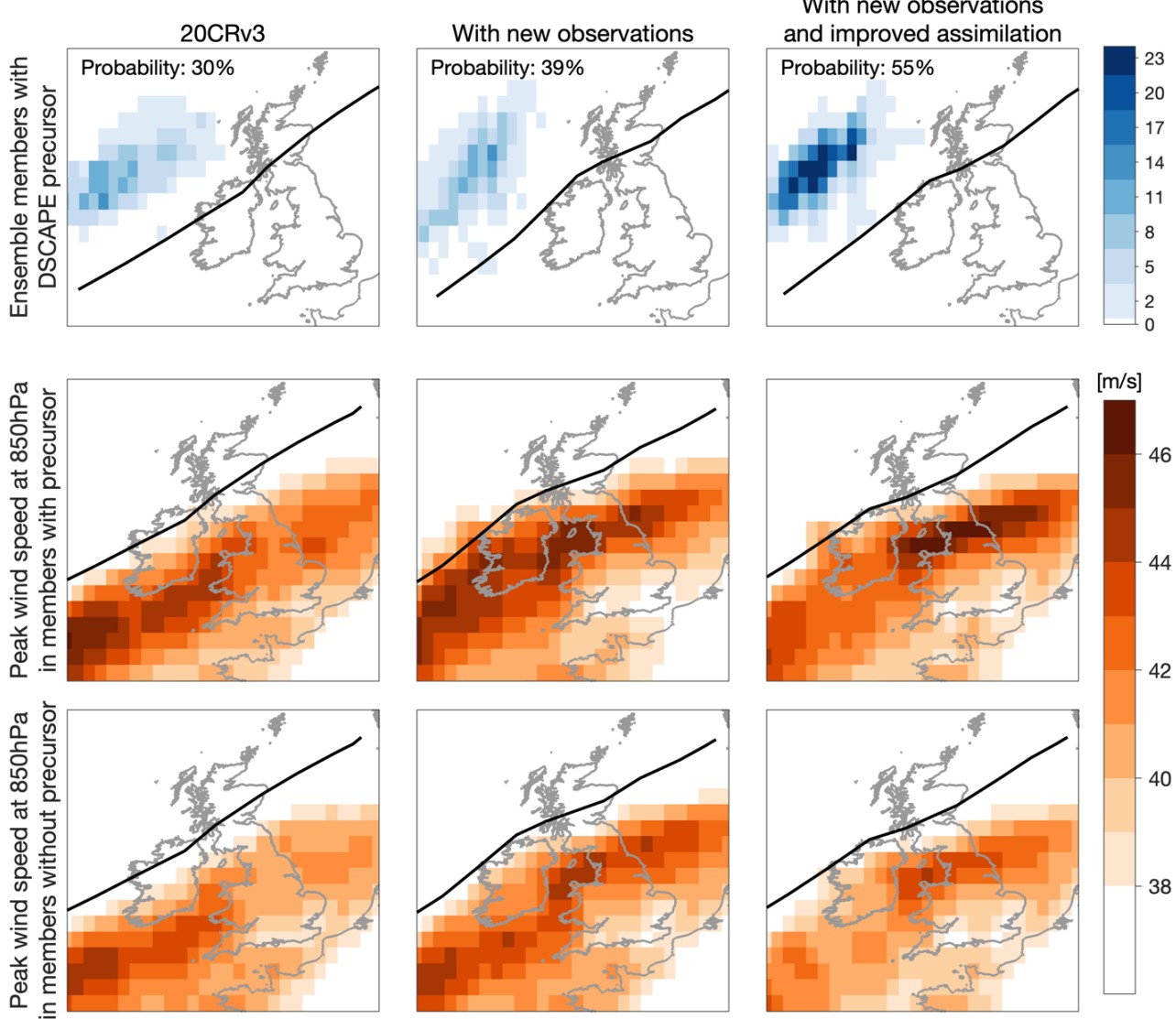

**Figure 6: DSCAPE metric during Storm Ulysses, showing the number of ensemble members with a sting jet precursor at that location at any point during the storm (top row). The number of members (out of 80) increases as the reanalysis is improved and the probability values signify the fraction of ensemble members with at least one grid point where the precursor is present at any time during the storm. The bottom two rows show the maximum wind speed at 850hPa in two sub-ensembles, using members with (middle row) and without (bottom row) a sting jet precursor. Peak winds >45m/s (100mph) are seen in the improved reanalysis at locations where known significant damage occurred over land, but only in the members with a precursor. The track of the minimum sea level pressure is shown by the black line in each panel, which varies slightly between reanalyses.**

## 4 Assessing rainfall variations during Storm Ulysses

An important test of the credibility of the Storm Ulysses reconstructions is to compare with additional independent data that are not assimilated into the reanalyses. North-west Europe, and the UK and Ireland in particular, have detailed instrumental observations that can be used for such an evaluation. In this case, both daily and even sub-daily rainfall observations can be independently compared with rainfall estimates generated within the reanalyses (see Appendix A for details). Figure S6 shows how

precipitation varies during the storm in all the original 20CRv3 and two experimental reanalyses.

Figure 7 compares the rainfall totals derived from interpolated in-situ observations (Hollis et al. 2019) and from the reanalysis for the two days of the storm and shows good agreement in the broad spatial patterns between 20CRv3 and the observations. None of the reanalysis experiments capture the large rainfall over the mountainous regions of the UK, presumably due to the coarse resolution of the

290 reanalysis compared to the small spatial scales of the steep orography. However, the reanalysis with additional observations and improved assimilation has drier conditions over central UK and wetter conditions over northern France than the original version, in even better agreement with the independent rainfall observations. The average ensemble spread in rainfall totals is reduced by 25% across the domain in this improved reanalysis compared to 20CRv3 (not shown).

It is also possible to compare higher frequency rainfall data. During this period the UK and Ireland had five meteorological observatories that were taking hourly observations of rainfall which can also be compared with the reanalysis. The blue bars in Figure 8 show these observations, integrated over the same 3-hour periods as the reanalysis. The grey and red lines show 20CRv3 and the improved reanalysis, highlighting that even on a sub-daily timescale, there is reasonable agreement with the

observations, both for the timing of the rainfall and the amounts. The exception is Fort William, which is in the most mountainous region of the UK, where the reanalysis is too low resolution to represent the variable orography. There is more rain in the observations than the reanalysis for this location (and also for Valentia to a lesser extent), however the timing of peak rainfall amounts is well represented. Although the ensemble mean rainfall does not change notably between 20CRv3 and the improved

reanalysis, the average ensemble spread across each location and 3-hour period is reduced by 24% in the improved version (not shown).

Slivinski et al (2021) demonstrated that interannual variability in rainfall is well represented in 20CRv3. This study extends those comparisons to an individual event. It provides evidence that 20CRv3 produces plausible estimates of rainfall during this extreme storm over most parts of the UK and

310 Ireland, and that this representation is further improved in the experiments performed. Further spatial downscaling would likely be required to reliably represent rainfall variations in mountainous regions.

# Rainfall during Storm Ulysses

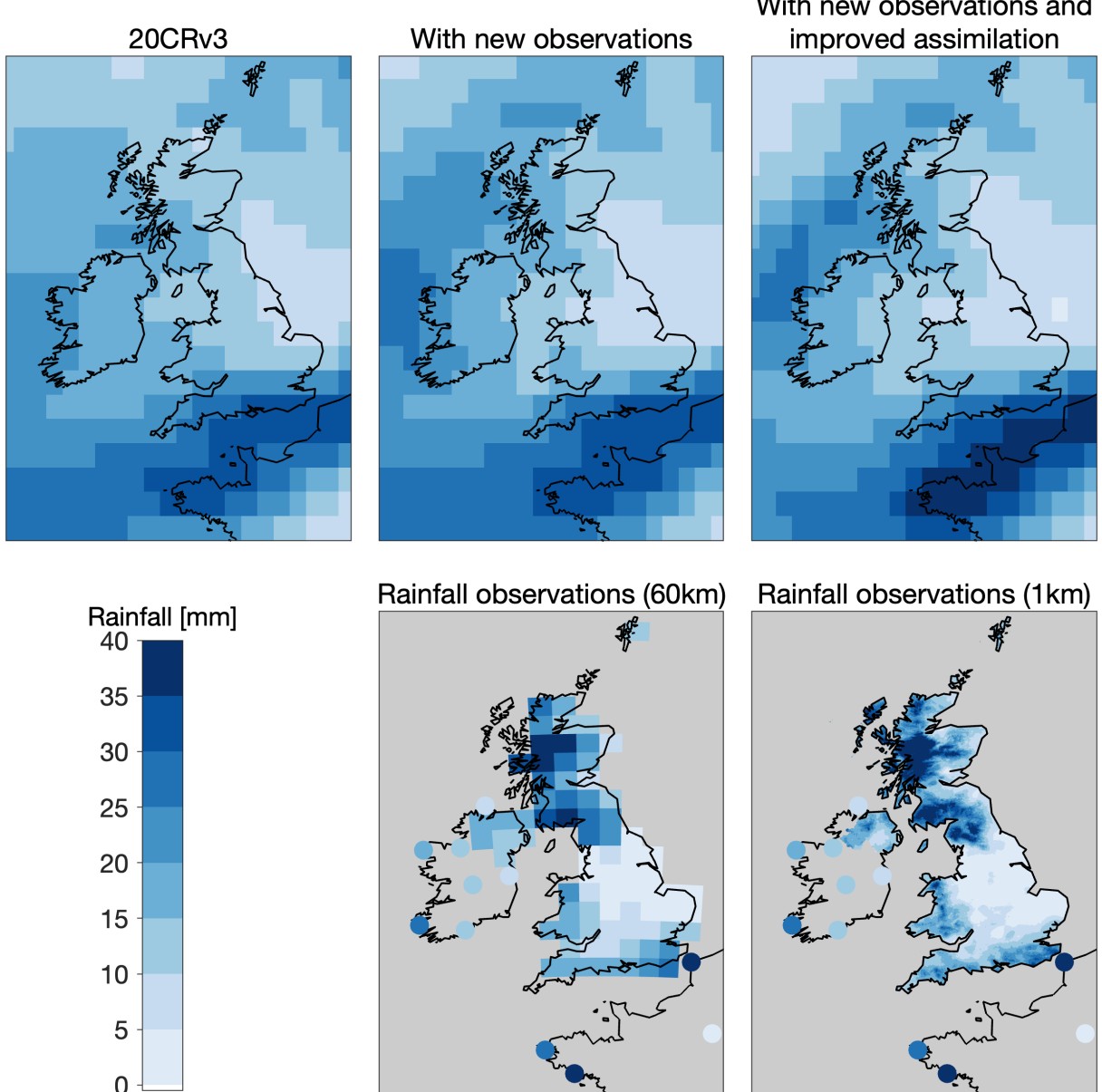

Figure 7: **Assessing rainfall variations during the storm. Rainfall in mm for the 48-hour period between 0900 UTC on 26th and 0900 UTC on 28th February 1903 in the three versions of the reanalysis (top row), compared with gridded rainfall reconstructions for the UK, interpolated from in-situ observations (HadUK-Grid, on two different spatial scales; bottom row). The 60km dataset roughly matches the spatial resolution of 20CRv3. Other available individual station rainfall observations for Ireland and France are shown with filled circles (see Appendix A). None of the rainfall data are assimilated in the reanalyses and so the observational data and reanalysis output are independent.**

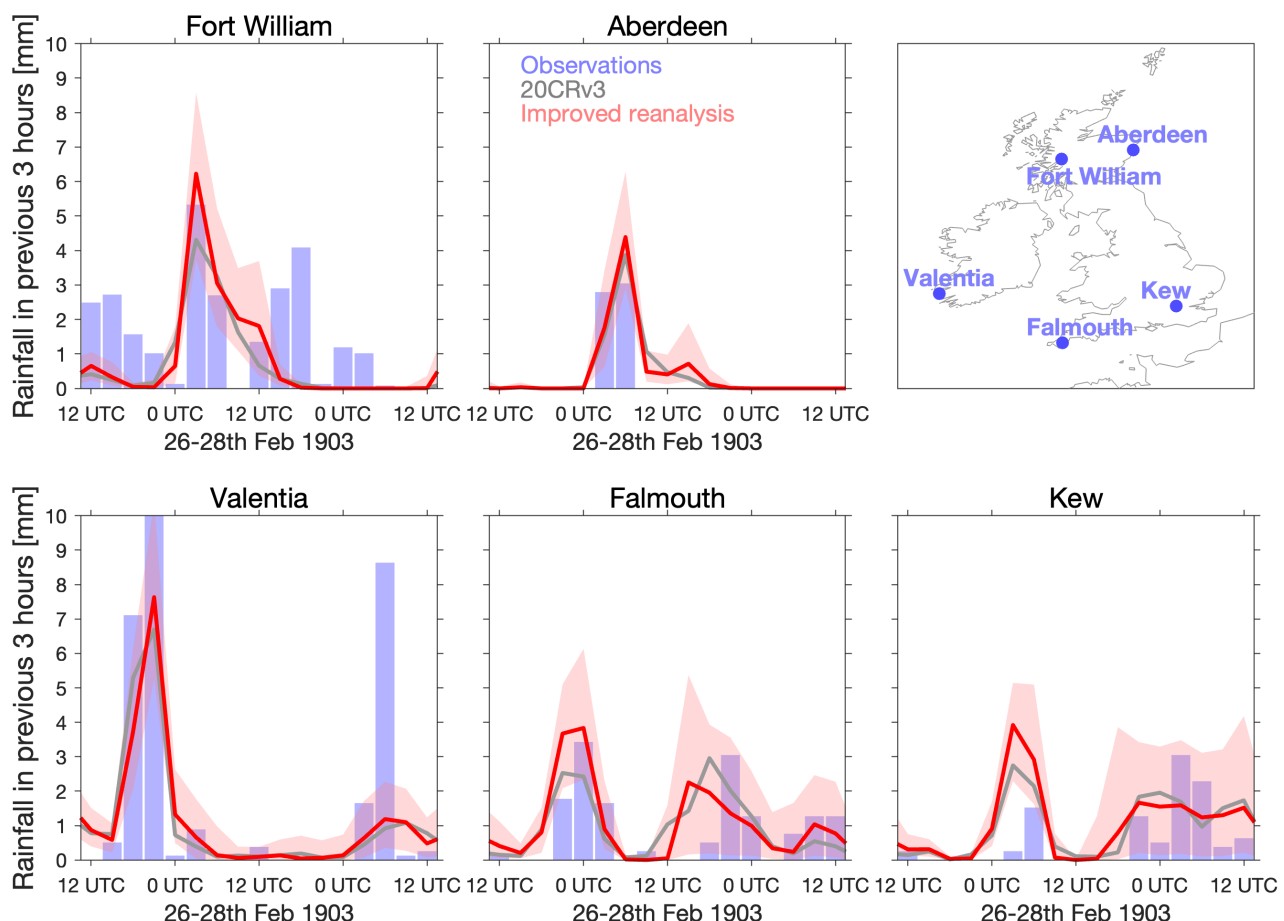

**Figure 8: Assessing high-frequency rainfall variations.** Rainfall every 3 hours during Storm Ulysses from 20CRv3 (ensemble median in grey) and the reanalysis with new observations and improved assimilation (ensemble median and 16-84% range in red) between 1200 UTC on 26[th] and 1200 UTC on 28[th] February 1903, for 5 locations across the British and Irish Isles. The 16-84% range is roughly equivalent to showing the ensemble standard deviation but is more appropriate for a non-normally distributed variable such as high-frequency rainfall amounts. These locations have hourly rainfall observations available which have been integrated over the same 3-hour periods (blue bars) as produced by the reanalysis. None of the rainfall data are assimilated in the reanalyses and so the observational data and reanalysis output are independent.

## 5 Assessing the coastal storm surge

Windstorms also produce coastal storm surges. We use the reanalysis winds and atmospheric pressure
fields to drive the UK Continental Shelf 3 (CS3) model, which is a hydrodynamic numerical ocean
model of the entire northwest European continental shelf with a resolution of approximately 12km.
These simulations produce estimates of storm surge heights around the British and Irish Isles (see
Appendix C for more details). This type of approach has previously been adopted globally (e.g., Muis et
al. 2016; Tadesse & Wahl 2021) and regionally (e.g., Haigh et al. 2014) using different reanalyses,
including for specific extreme events (Choi et al. 2018, Meyer et al. 2022) with mixed success.

Figures 9a,b show maps of the height of the simulated maximum storm surge during Storm Ulysses
using 20CRv3 and the reanalysis with additional observations and improved assimilation. More
precisely, the non-tidal residual is shown which indicates the difference between the water height driven
by the meteorological forcing after the astronomical tidal component has been removed. The improved
reanalysis fields drive a larger storm surge on the north-west coast of England and around Irish coasts,
and a smaller storm surge in other locations (Figure 9c), with considerably reduced uncertainty. This is
consistent with the pattern of stronger winds shown in Figure 3.

Figures 9d,e also compare the simulated storm surge with the same metric derived from high-frequency
tide gauge observations for two sites near Liverpool (within 15km of eachother). The tide gauge data
has recently been digitised from paper archives in another citizen science project (see Appendix A).
These two sites are close to the region of strongest winds during Storm Ulysses and the peak simulated
storm surge. The improved reanalysis fields drive a larger surge (by 0.35m) than the original reanalysis,
in better agreement with the observations. However, the observed storm surge (around 2.5m) is still
slightly larger than the simulations, hinting that the reanalysis might still be underestimating the wind
strength, and could be further improved through, for example, improved resolution or addition of more
pressure observations to better constrain the wind fields. Alternatively, the coastal surge model could be
slightly underestimating the local response to the winds and increased spatial resolution may help
resolve this. There is also more variability in the observations than the reanalysis, but this is not
unexpected due to complex local tidal features in this region. This verification against independent data
is encouraging for the credibility of the reanalysis, and for using the reanalysis to estimate storm surges
at other locations and time periods where tide gauge data are not available.

There are no reports of flooding in Liverpool during this storm because the maximum surge occurred
during neap tides, and not at high tide. As a result, the skew surge (peak observed height minus peak
predicted height during the tidal cycle) was around 1.2m. Overall, the storm surge is one of the ten
largest observed events between 1857-1903 (the period of data recently rescued) and is larger than any
observed event in the available modern Liverpool tidal records (1991-2021). This suggests that this
storm surge was a roughly once-per-decade event and would likely have caused flooding if the peak
surge had occurred during high spring tides. Improved knowledge of such events will inform risk
estimates of coastal flooding, especially as sea levels in this region have increased by around 0.2m since
this storm.

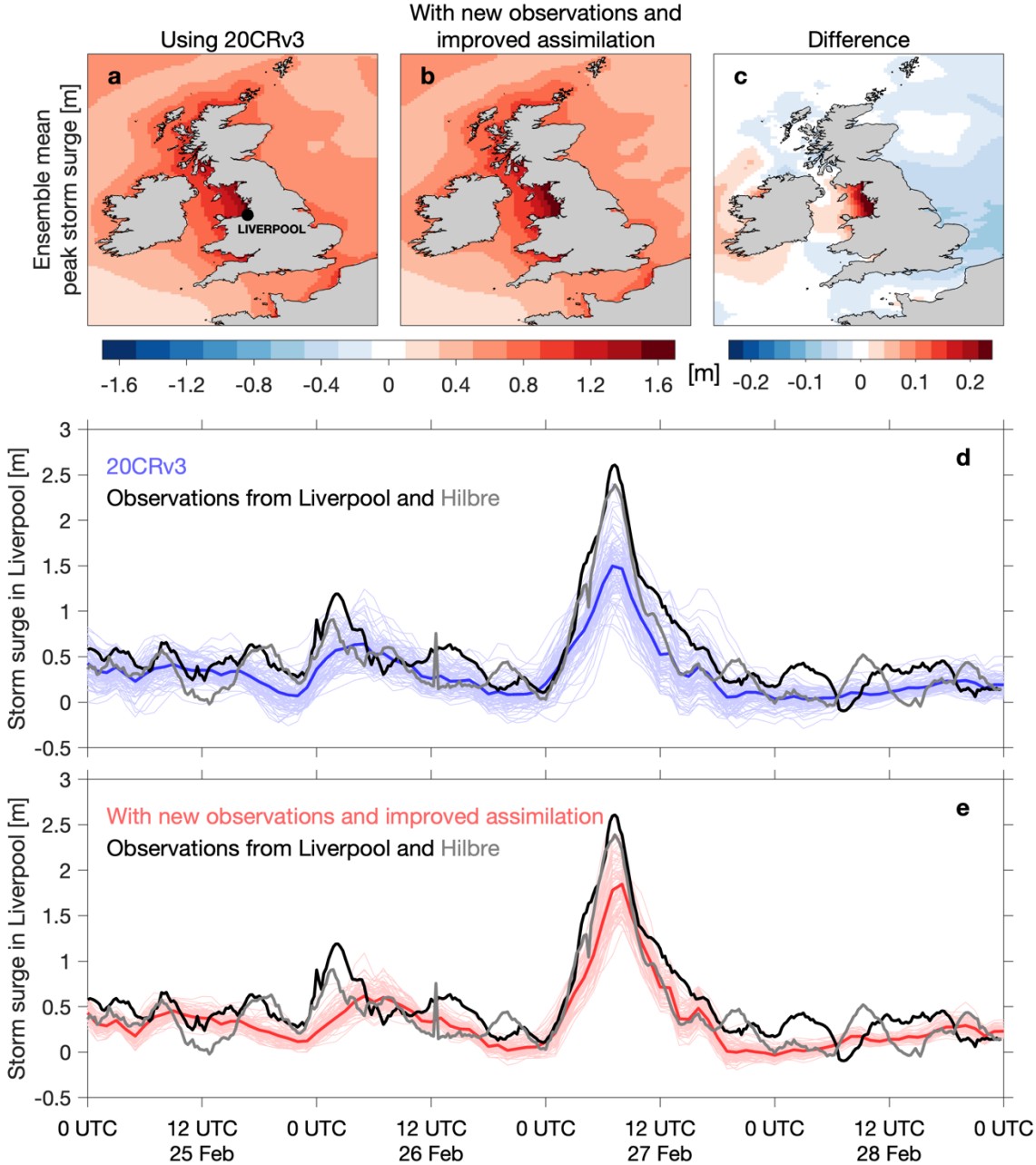

**Figure 9: Storm surge simulations for Storm Ulysses.** Top: maps of the ensemble mean of the maximum storm surge during Storm Ulysses, for two versions of the reanalysis (a,b), and the difference ©. Bottom: simulated storm surge height at Liverpool for each ensemble member (thin lines) and ensemble mean (thick lines) using two versions of the reanalysis (d,e), compared with the newly rescued tide gauge observations for Liverpool Docks and Hilbre Island, for 25th-28th February 1903.

## 6 Benefits of rescuing observations for improving reanalyses and estimating risk

The recovery of historical weather observations from paper archives is informing our knowledge and understanding of the risks from extreme weather. Any approach to estimating risk by identifying plausible worst-case outcomes (Thompson et al. 2017) or developing storylines of severe weather events (Shepherd et al. 2018) would benefit from longer sampling of real-world behaviour and improved historical knowledge (Woo & Johnson 2018; Pinto et al. 2019).

As an example, during the modern period (1950-2015) the maximum instantaneous wind speed (based on 3-hourly data) experienced at a grid point in northern England (54.4N, 2.0W) was 21.2m/s in 20CRv3. During Storm Ulysses, this location was in the region of peak winds over land and experienced a wind speed of 22.0m/s. The modern period data suggests that the unprecedented winds experienced during Storm Ulysses would be rarer than once-in-100 years for that location. Having a credible reconstruction for such a rare event provides valuable information on plausible risks and potential damage. Note that these quoted wind speeds will be substantially less than sustained wind speeds or gusts, motivating future downscaling of this storm to better quantify the extreme nature of the winds.

This end-to-end case study demonstrates how combining modern weather forecasting and data assimilation techniques with measurements of surface pressure taken over 100 years ago can credibly reconstruct details of one of the most severe European windstorms in the instrumental period, providing support for the capability of this reanalysis approach to reconstruct extreme events in general. This study is also a clear example of how the addition of newly-rescued meteorological and related climatological observations can directly improve reanalyses of such extreme events, and provide independent validation of the reconstruction.

Comprehensive rescue of existing weather observations, stored on paper in various archives, would allow much more precise and accurate reconstructions of many other similar events across Europe (and any region with enough data available), including other extreme weather events such as heatwaves and floods (Brönnimann et al. 2018). Better knowledge and understanding of these historical extreme events would allow observed trends in extreme events to be put into a longer-term context and help identify where present-day and future risks have been underestimated because such extreme events may not have yet been observed during the modern period.

## Appendix A   Additional observations

The additional pressure and rainfall observations used come from a range of sources. The largest component comes from the Weather Rescue citizen science project (Craig & Hawkins, 2020) which digitised 11 years of the UK Met Office 'Daily Weather Reports' (1900-1910; e.g. Figure A1). These reports include twice-daily surface pressure observations and daily rainfall amounts from 57 locations across the UK, Ireland, and north-west Europe. The new pressure data also includes hourly observations taken on the summit of Ben Nevis in Scotland and in the nearby town of Fort William, which were also

transcribed by volunteers (Hawkins et al. 2019). The final source of data is 19 Second Order Stations and eleven locations with Climatological Returns in the UK Met Office digital archives (NMLA, 2023)
which were additionally transcribed for the period around Storm Ulysses, with two pressure observations per day. Note that in 1903, when Storm Ulysses occurred, Ireland was not an independent country and was part of the UK. This can be seen in Figure A1 which describes observations from locations in present-day Ireland as being part of the 'British Islands'. We have used present-day national boundaries when describing locations in the text and use 'British and Irish Isles' as a more inclusive
term where appropriate. Pressure observations from tens of more locations are potentially available for the British and Irish Isles (and other countries) for this event, including hourly data from several sites, but these have not yet been digitised from the paper archives.

In 20CRv3, the land stations are all assigned an uncertainty of 1.2hPa (surface pressure) or 1.6hPa (sea level pressure), and ship observations are assigned an uncertainty of 2.0hPa. This is unchanged in our
experiments. The added observations mainly come from locations that were regularly inspected by the Met Office, suggesting the data is of high quality and perhaps the uncertainty assigned is too large.

The gridded daily rainfall data in Figure 7 is from the HadUK-Grid dataset (Hollis et al. 2019), summed over the two days of Storm Ulysses. The individual station observations in Figure 7 are taken from the Daily Weather Reports and the Second Order Station books. The hourly rainfall observations in Figure
8 were digitised manually just for this event from the Hourly Books for the four Met Office Observatories (Falmouth, Kew, Aberdeen and Valentia), stored in the Met Office Archives. Hawkins et al. (2019) contains the hourly rainfall data for Fort William.

Two tide gauge series with 15 minute resolution covering the Ulysses storm were recently recovered from scanned paper records by the UK Tides citizen science project (Williams & Matthews, in prep.).
The two nearby sites are Liverpool Docks (53.4052 ºN, 2.9985 ºW) and Hilbre Island (53.3851 ºN, 3.2293 ºW). The precision of the tide gauge data is 1 inch, and the data for the period from 25th-28th February 1903 are available (see Figure A1 for an example of the scanned pages) and have been screened for quality-control. The errors are hard to constrain without contemporary levelling information, but the high coherence of the Liverpool and Hilbre data suggests they are within a few tens
of cm. The tidal data for the whole period rescued (1857-1903) have not yet been fully quality-controlled so we cannot yet be precise about how extreme the Ulysses storm surge was within the full context of the longer period, but it is certainly in the top-10 events. There are no other digitised high-frequency tide gauge records for the UK for this period, although several do exist on paper.

Figure A1: The UK Daily Weather Report observations page for 27th February 1903 (left; from NMLA (2023)) and tide gauge measurements from Liverpool Docks for 26th-27th February 1903 (right; supplied by authors).

## Appendix B   Improvements to data assimilation scheme

We used the openly available data of the 20CRv3 reanalysis for this storm (Slivinski et al. 2019b) and performed two additional experiments with the same reanalysis system. The first experiment was
445 identical to the original except for assimilating thousands of additional newly rescued surface pressure observations. The second experiment repeated the first experiment with a change to the data assimilation scheme.

In the 20th Century Reanalysis assimilation system, the background (prior) fields are provided from the underlying numerical weather prediction model with prescribed pentad sea surface temperatures
(interpolated to daily), monthly sea ice concentration, and monthly radiative forcing. Surface pressure observations are assimilated with an ensemble Kalman filter (Whitaker and Hamill 2002) to generate the reanalysis. One common issue with ensemble filters is so-called "ensemble collapse", in which the ensemble spread can collapse to 0 in sequential assimilation cycles (Anderson & Anderson 1999; Whitaker and Hamill 2002). Generally, ad-hoc inflation methods are needed to increase the ensemble
spread. In the 20CRv3 system, the inflation method is relaxation-to-prior-spread (RTPS; Whitaker and

Hamill 2012; Slivinski et al 2019a), where the analysis ensemble spread is "relaxed" back to the prior ensemble spread by a temporally- and spatially-varying parameter λ. This parameter depends on the observation network density at that time and location, as well as a hyperparameter $p_{relax}$:

$$\lambda_{\text{inf}(x,y,t)} = \frac{p_{relax}\left(\sigma_{b(x,y,t)} - \sigma_{a(x,y,t)}\right)}{\sigma_{a(x,y,t)}} + 1$$

where $\left(\sigma_{b(x,y,t)}\right)$ is the standard deviation of the background ensemble and $\left(\sigma_{a(x,y,t)}\right)$ is the standard deviation of the analysis ensemble before inflation. The hyperparameter $p_{relax}$ can vary from 0 to 1 and determines the sensitivity of λ to the observation density: the higher $p_{relax}$ is, the more the analysis ensemble can be relaxed back to the prior ensemble. Essentially, the inflation is increased with high observation density, and decreased with low observation density (see figure 3 of Slivinski et al 2019a).

However, $p_{relax}$ itself needs to be tuned; due to computational cost, effort, and number of parameters that need to be tuned, only a few initial tests were completed, resulting in $p_{relax} = 0.9$ in the northern hemisphere, 0.7 in the southern hemisphere, and a linear transition between the two values in the tropics. However, results from these experiments (see below) suggest that further tuning could be beneficial, since the addition of many new observations did not have as large of an impact on the

analysis mean or spread as expected. Therefore, the subsequent experiment with 'improved assimilation' was run with the new observations, as well as decreasing $p_{relax}$ to 0.5 everywhere. This ultimately prevented the analysis ensemble spread from being relaxed back to the prior ensemble spread as much, effectively allowing the observations to have a stronger impact, as shown in Figure 3.

We consider whether the changes to RTPS improve the reanalysis by comparing with independent

observations. Ideally the ensemble spread of the reanalysis should be 'reliable', i.e., it appropriately represents the uncertainty given the available observations. This reliability can be tested by comparing with pressure observations that are not assimilated. For the period of Storm Ulysses, we have pressure observations from five additional locations that were withheld from the reanalysis (see right hand panel of Figure B1). These locations were chosen to cover a wide range of locations across the UK and

Ireland. The data were digitised manually for this event, except for Durham which recently became available for a longer period (Burt 2021).

We have extracted the reanalysis ensemble mean and ensemble spread from the locations and times of these independent observations and performed a mean bias correction. A bias correction is also included within the reanalysis assimilation cycle so this approach roughly mimics the reanalysis approach. This

process allows a root mean square error (RMSE) between the observations and reanalysis, and a mean ensemble spread for the times of the observations, to be calculated for each location for February 1903 (Figure B1). A reliable ensemble would show similar values for RMSE and ensemble spread.

For the original reanalysis, the ensemble spread is much larger than the RMSE for each location (blue symbols are to the right of the 1:1 dashed line), suggesting that the ensemble is under-confident (or

over-dispersive) in the atmospheric circulation patterns. In the experiments with additional observations (red symbols), the ensemble spread and RMSE have both been reduced as expected, but the ensemble

remains under-confident. In the experiment where the RTPS parameter is reduced (yellow symbols), the RMSE and ensemble spread now lie closer to the 1:1 line, indicating a more reliable ensemble.

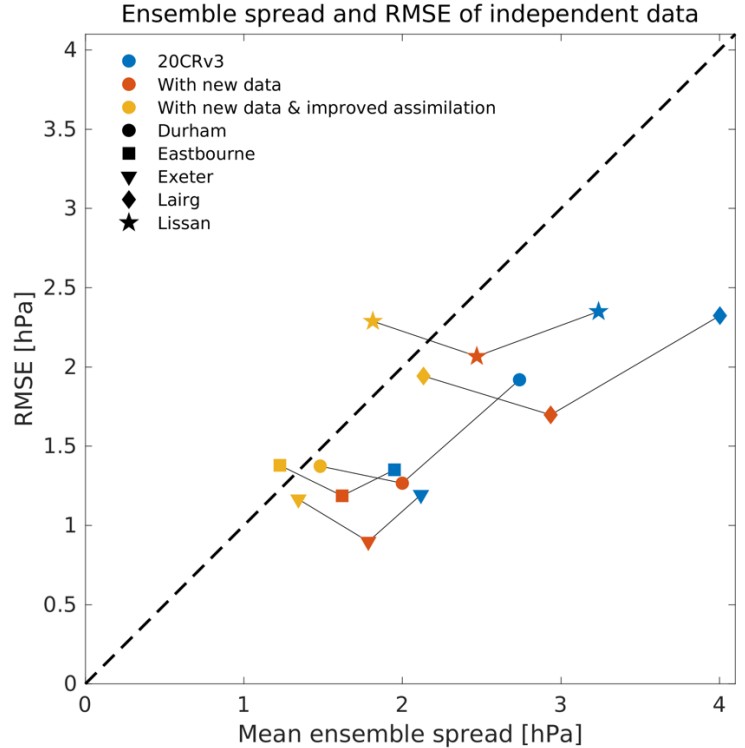 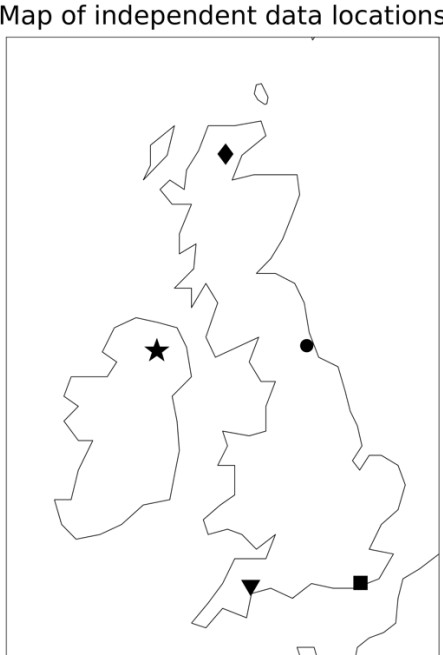

 **Figure B1: Ensemble reliability. Mean ensemble spread of the reanalysis, and RMSE of the reanalysis compared to unassimilated independent observations, for five locations (shapes) during February 1903. Different versions of the reanalysis are shown with different colours. None of these observations are assimilated in any version of the reanalysis. Adding the additional observations reduces both the ensemble spread and the RMSE (moving from blue to red symbols), and the improved assimilation has made the reanalysis more reliable (moving from red to yellow symbols, RTPS reduced from 0.9 to 0.5) with a small increase in RMSE. The**
 **dashed line represents 'perfect' ensemble reliability, when RMSE and ensemble spread are equal.**

Although these results are encouraging that a smaller RTPS parameter is producing a more reliable ensemble, if also accounting for observational uncertainty, the ensemble spread should be slightly smaller than the RMSE, i.e. the plotted points should fall to the left of the 1:1 line. It is therefore likely that the ensemble is still slightly under-confident even with the reduced RTPS.

 Further experiments for a much longer period would be required to rigorously assess the ensemble, likely including examining other parameters such as the assumed uncertainty in each observation.

We also consider the modern period with a similar set of tests. There is a possibility that the RTPS parameter in the original 20CRv3 might also need to be reduced for the modern period which would make the comparison of wind ranks in Figure 4 potentially unfair. However, Figure B2 highlights that
 for two example years (1953 and 2003, i.e. 50 and 100 years after Storm Ulysses), the original 20CRv3

is roughly reliable when compared to unassimilated pressure data from one location (Reading, UK, 51.5ºN, 1.0ºW) which is now available for each year in the comparison. The Reading data were assimilated in the two reanalysis experiments, so they are not included in Figure B1.

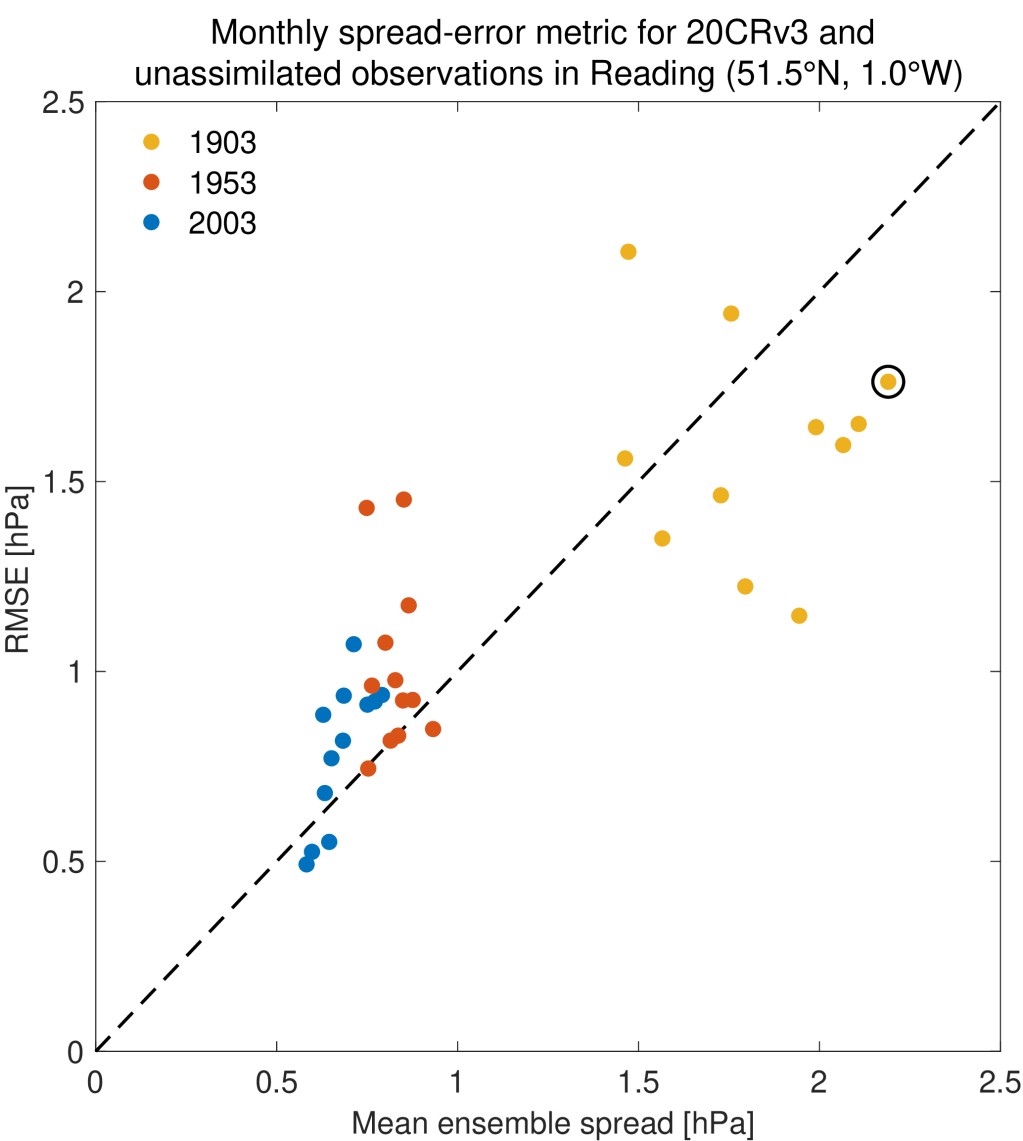

**Figure B2: Ensemble reliability across years.** Mean ensemble spread of the reanalysis, and RMSE of the reanalysis (both in hPa) compared to unassimilated independent observations from one location where data is available for an extended period (one dot per month from three different years). For the modern period (using 1953 and 2003 examples), the ensemble is roughly reliable so there is no evidence that the RTPS parameter needs to be altered for that time period. The open circle indicates February 1903.

## Appendix C   Storm surge model

To model the storm surge and tide we used the UK Continental Shelf 3 (CS3) model. This model was developed at the National Oceanography Centre in the UK and is based on a finite-difference discretisation of the fully non-linear, depth-averaged Navier-Stokes equations (Proctor and Flather, 1983; Flather et al. 1991). CS3 was extensively used for operational forecasting by the Met Office from 1991 to 2006 and is one of the most validated operational storm surge forecasting models in the world. The model covers the entire northwest European continental shelf on a 1/9° latitude by 1/6° longitude grid, giving a resolution of approximately 12km. We applied tidal forcing at the open lateral boundaries using the 15 largest constituents derived from a harmonic analysis of a larger area ocean model. Wind stress was calculated using a quadratic stress law where the drag coefficient is derived from observations using the parameterisation of Smith and Banke (1975). We ran the hydrodynamic model 160 times from the start of 25th February to the end of 28th February 1903 simulating total water level (e.g., tides plus storm surges) using wind (u and v components) and atmospheric pressure fields from each of the original 80 reanalysis ensemble members and then for the 80 improved ensemble members. The reanalysis produces 3-hourly winds and pressure fields, which were interpolated to hourly for the simulations. We also ran an additional tide-only simulation. We subtract the predicted astronomical tidal heights from each of the total water level simulations to estimate the storm surge components. We save model results for each model grid cell every 10 minutes and calculate maps of the maximum storm surge over the event for each original and improved ensemble member. We also extracted the storm surge time-series at the nearest point to the Liverpool tide gauges.

## Data availability

The complete 20th Century Reanalysis dataset is openly available (https://portal.nersc.gov/project/20C_Reanalysis/). For the short period around Storm Ulysses, the data from 20CRv3 and the reanalysis experiments performed are available here: https://github.com/ed-hawkins/ulysses-storm-data. The additional observations used are available on Zenodo with doi 10.5281/zenodo.7765124.

## Author contributions

EH conceived and led the project and analysis, with contributions from all co-authors. PB, GPC, HH, CM and LS assisted with the design and running of the reanalysis experiments. KK and IDH carried out the storm surge experiments. SNB and SB assisted with data recovery, and JW provided the tidal data. SLG and OMA provided guidance on the sting jet analysis. EH prepared the manuscript with contributions from all co-authors.

## Competing interests

None

## Acknowledgements

We thank the observers who took the original weather observations over a century ago, those who collated the data so carefully at the time, and the archivists who have preserved the paper material ever since. We also gratefully acknowledge the thousands of citizen scientist volunteers who gave their spare time to help digitise and recover the weather and tide gauge observations used, and Zooniverse.org for providing the citizen science platform. We also thank Andy Matthews and Elizabeth Bradshaw for the

retrieval of Liverpool and Hilbre tide gauge data. Support for the Twentieth Century Reanalysis Project version 3 dataset is provided by the U.S. Department of Energy, Office of Science Biological and Environmental Research (BER), by the National Oceanic and Atmospheric Administration Climate Program Office, and by the NOAA Physical Sciences Laboratory. This research used resources of the National Energy Research Scientific Computing Center (NERSC), a U.S. Department of Energy Office

of Science User Facility located at Lawrence Berkeley National Laboratory, operated under Contract No. DE-AC02-05CH11231. EH was supported by the National Centre for Atmospheric Science. EH and APS were supported by the NERC GloSAT project. PB was supported by the Met Office Hadley Centre Climate Programme funded by BEIS. GPC, LS, and CM are supported in part by NOAA cooperative agreement NA22OAR4320151, by the NOAA Climate Program Office and NOAA

Physical Sciences Laboratory.

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
