# Peer review of "Rescuing historical weather observations improves quantification of severe windstorm risks"

_EGUsphere, 2022_

## Referee Comment (RC1)

**Review of *'Rescuing historical weather observations improves quantification of severe windstorm risks'* by Hawkins et al.**

The authors present an overview of Storm Ulysses and through the use of historical data records are able to produce new reconstructions of the storm using historical reanalysis. This new reconstruction results in a more intense storm and the authors highlight the major attraction of rescuing historical data for the purpose of expanding the record of historical storms, especially those of high intensity. In general, this manuscript is very well written and the authors clearly demonstrate the benefits of this sort of analysis and utilising historical written records for improving our understanding of historical storms. I only have several minor points of issue with this manuscript and after being addressed I would have no issues recommending this be accepted for publication. My main issue is around the framing of this manuscript and I believe slightly more background/justification for the analysis is required. This, and my other minor comments are detailed below.

**Major comment**

1. I believe a broader introduction is required and motivation of performing these re-runs is required. This is noted clearly in the abstract that a transformation of our understanding of historical variability is possible, but the authors do not note this until L141-146. No real aim or scientific purpose of the study is given until the results are discussed and this is something that needs to be rectified.

**Minor comments**

1. For all figures I would recommend labelling of panels via a, b, c, d, etc. This would make it easier to know which panels are being referred to rather than 'bottom row', etc.
2. L125, please include figure reference.
3. L125, Fig 4. How does the assimilation density of 20CRv3 compare to the assimilation performed by the authors using the new data? WIll this affect the ranking that you have fone in Fig. 4 (d-f)? By improving the density of Ulysses this may mean it's winds are not representative of the 1950-2015 reanalysis and so the ranking may not be correct. Please clarify this.
4. Fig 4 (and throughout). It may be useful to show the plots of new data (and improved DA) as difference plots to highlight exactly where the re-imagined storms have strengthened relative to 20CRv3.
5. L221-224, how do the quoted percentages of ensemble members (49% and 22%) relate to the probabilities quoted in Fig 6. These values are different and I find it hard to understand why or how the authors have computed them to be different. This needs clarifying.
6. L226-228, this feels like repetition of two paragraphs prior. Please consider re-phrasing.
7. L231-232, Fig 6, it would be good to also quote the windspeeds of the non-precursor members. Furthermore, are there any statistical differences in the distribution of windspeeds simulated between the precursor and non-precursor members? If not it needs to be stated that even though the ensemble mean is higher, there is no statistical increase in simulated windspeed with sting jet precursors.

8. L316-319, Fig 8, Is the gauge data used in Fig, 8 a point estimate? If so I would not expect the output of the coastal surge model to match that of a point estimate as it has resolution of 12km. It may just be that the coarse nature of the reanalysis is unable to simulate such wave heights. This section is stated as if the storm is still not simulated to the correct strength is the driving factor of this, whereas it should be restated (in my opinion) that the difference in resolution of the two datasets is the leading driver of the difference and that an underestimation in intensity may be another reason why.

---

## Referee Comment (RC2)

*Review:* https://doi.org/10.5194/egusphere-2022-1045
**Rescuing historical weather observations improves quantification of severe windstorm risks**
Hawkins et al.

This study shows the improvement of simulations of a historical intense windstorm due to added rescued observations from the time. The authors compare a reanalysis with added observations and new data + improved data assimilation. It is shown that the spread decreases and distinct features appear in the ensemble mean with more observations. The study is really interesting, and it clearly shows the value of rescued weather data. I have only minor concerns, which should be discussed more thoroughly.

**Major comments:**

- I certainly agree that rescued observations are of great value for improving simulations of historic weather events. However, I feel the positive statements from comparisons and argumentations could be toned down a bit at times and more discussion is needed. For example, a short discussion about their trustworthiness, accuracy, error range of the rescued observations should be included.
- The discussion is purely based on the ensemble mean (except for the sting jet precursor). I wonder if the missing wind jets in Figure 5 and that you discuss in l. 164ff are rather due to looking at the mean and are actually present in individual ensemble members. With a large spread, the maximum wind speeds of smaller features, such as the cold conveyor belt jet or sting jet, probably differ in location and, hence, are weaker in the ensemble mean. Of course, Figure 5 shows an improvement nonetheless, however, the argumentation why that is changes and you should state that the features "are not present in the 20CRv3 mean" (l. 172).

**Minor comments:**

- Please consider using hPa instead of mb.
- How did you track the storm?
- Please be consistent with figure labels (e.g., "new data" vs. "new observations", colorbar labels).
- Some figures are not discussed to their full extent. You show interesting information in the figures, which are – sometimes – not even mentioned in the text (e.g., probabilities in Fig. 6)

- l. 72f: As you state, 960mb is an estimate, so the comparison of the pressure minima in simulations with this value should be put in relation and not seen as the absolute truth.
- l. 85ff: Please add 1-2 sentences with more information about the added data and especially the improved data assimilation to the main text. It would be good to at least have an idea about the improvements without having to read the Appendix, which should then be for readers with further interest.
- Figure 3 and elsewhere: Please consider putting the colorbar labels right next to the colorbar.
- l. 218: Please elaborate the "simpler grid point approach". Do you mean you simply make the tool independent of neighbouring grid points, hence you could use it on every grid point independently? Please discuss shortcomings of this approach.

- l. 221ff: When do you define a member to show precursors: Is this already the case for only one grid point? How do these percentages compare to the probabilities in Fig. 6?
- Figure 6: The difference between 20CRv3 and new data seems to be much smaller than new data and new data + improved assimilation. Can you comment on this? Could the improved assimilation be more important for the improvement than the new observations? However, this is not really the case in other figures.
- l. 257: "observed": As in the caption, you should at least mention the HadUK-Grid, i.e., interpolated in-situ observations.
- Figure 8: Please consider another colour scheme. Furthermore, what is the reasoning behind the 16-84% range?

---

## Author Comment (AC1)

**Response to Reviewer 1:**

We thank the reviewer for their careful reading of the manuscript and their thoughtful comments which we discuss below.

*I believe a broader introduction is required and motivation of performing these reruns is required. This is noted clearly in the abstract that a transformation of our understanding of historical variability is possible, but the authors do not note this until L141-146. No real aim or scientific purpose of the study is given until the results are discussed and this is something that needs to be rectified.*

We agree with the reviewer and have added some additional discussion in the opening section.

1. *For all figures I would recommend labelling of panels via a, b, c, d, etc. This would make it easier to know which panels are being referred to rather than 'bottom row', etc.*

We have added panel labels in some of the figures.

2. *L125, please include figure reference.*

We have added an extra reference to Figure 4 in this paragraph.

3. *L125, Fig 4. How does the assimilation density of 20CRv3 compare to the assimilation performed by the authors using the new data? Will this affect the ranking that you have done in Fig. 4 (d-f)? By improving the density of Ulysses this may mean its winds are not representative of the 1950-2015 reanalysis and so the ranking may not be correct. Please clarify this.*

We understand the reviewer's concern and is the reason why we included Figure B2 in the original manuscript and the discussion in L467-473. The density of observations assimilated in the experiments for 1903 is much smaller than for the modern period (1960s onwards), even after the addition of extra data. When compared to the 1950s, the experiments with added observations for 1903 have more locations over the UK, although the observation frequency is often lower. Figure B2 and the associated text highlights that the ensemble spread appears roughly reliable in the modern era when using one set of available independent observations and two example years. Although this is a simple test, it appears as though the assimilation scheme change might not be required for the 1950-2015 period and so the comparison of wind ranks is considered to be fair.

4. *Fig 4 (and throughout). It may be useful to show the plots of new data (and improved DA) as difference plots to highlight exactly where the re-imagined storms have strengthened relative to 20CRv3.*

We have considered this issue and decided to retain the figures as before as the main differences are clearly visible and it is the absolute values that are being assessed with the independent data.

5. *L221-224, how do the quoted percentages of ensemble members (49% and 22%) relate to the probabilities quoted in Fig 6. These values are different and I find it hard to understand why or how the authors have computed them to be different. This needs clarifying.*

Thanks. This was an error in the text which had not been fixed. The percentages now match between the figure and the text.

> 6. *L226-228, this feels like repetition of two paragraphs prior. Please consider rephrasing.*

The text has been edited to be less repetitive.

> 7. *L231-232, Fig 6, it would be good to also quote the windspeeds of the non-precursor members. Furthermore, are there any statistical differences in the distribution of windspeeds simulated between the precursor and non-precursor members? If not it needs to be stated that even though the ensemble mean is higher, there is no statistical increase in simulated windspeed with sting jet precursors.*

The wind speeds in the members without a precursor are significantly lower than the members with a precursor and this is now added.

> 8. *L316-319, Fig 8, Is the gauge data used in Fig. 8 a point estimate? If so I would not expect the output of the coastal surge model to match that of a point estimate as it has resolution of 12km. It may just be that the coarse nature of the reanalysis is unable to simulate such wave heights. This section is stated as if the storm is still not simulated to the correct strength is the driving factor of this, whereas it should be restated (in my opinion) that the difference in resolution of the two datasets is the leading driver of the difference and that an underestimation in intensity may be another reason why.*

Experience with present day records and operational storm modelling suggests that the underprediction of the storm intensity, and precise timing and direction of wind fields, is most likely to be the cause of the discrepancy, which is quite small. Whilst the data are point estimates, the gauges were designed to physically smooth local wave action over several minutes, as is done (slightly differently) by a modern stilling well. The signals at Liverpool and Hilbre, which are about 10 miles apart along the coast, are closer to each other than to the model at the peak surge. However, L320 has been edited to say that improved resolution of the storm surge model might help resolve remaining discrepancies.

---

## Author Comment (AC2)

**Response to Reviewer 2:**

We thank the reviewer for their careful reading of the manuscript and their thoughtful comments which we discuss below.

- *I certainly agree that rescued observations are of great value for improving simulations of historic weather events. However, I feel the positive statements from comparisons and argumentations could be toned down a bit at times and more discussion is needed. For example, a short discussion about their trustworthiness, accuracy, error range of the rescued observations should be included.*

In the 20CRv3 system the land stations are all assigned an uncertainty of 1.2mb (surface pressure) or 1.6mb (sea level pressure), and ship observations are assigned an uncertainty of 2.0mb. This is unchanged in our experiments. The cited Craig & Hawkins (2020) paper discusses the collection and QC of most of the new pressure data, but we now include more details about the number of stations and their assumed uncertainty in Appendix A. It is likely that, as these stations were formal observatories that were regularly inspected by the Met Office, the data is of very high quality and perhaps the uncertainty assigned is too large. We have made some edits and deletions to slightly tone down the text in some places, such as Section 3.

- *The discussion is purely based on the ensemble mean (except for the sting jet precursor). I wonder if the missing wind jets in Figure 5 and that you discuss in l. 164ff are rather due to looking at the mean and are actually present in individual ensemble members. With a large spread, the maximum wind speeds of smaller features, such as the cold conveyor belt jet or sting jet, probably differ in location and, hence, are weaker in the ensemble mean. Of course, Figure 5 shows an improvement nonetheless, however, the argumentation why that is changes and you should state that the features "are not present in the 20CRv3 mean" (l. 172).*

We agree and have clarified that the features are not present in the ensemble mean of 20CRv3 (old line 172). None of the ensemble members of 20CRv3 have a wind jet.

- *Please consider using hPa instead of mb.*

Thanks – we have considered this and prefer to stick to mb.

- *How did you track the storm?*

The storm is tracked by interpolating the gridded pressure field to find the local minimum. This is added to the text.

- *Please be consistent with figure labels (e.g., "new data" vs. "new observations", colorbar labels).*

Thanks for spotting this - we have changed all the figure panel titles to say 'new observations'.

- *Some figures are not discussed to their full extent. You show interesting information in the figures, which are – sometimes – not even mentioned in the text (e.g., probabilities in Fig. 6)*

Thanks. Some additional text has been added discussing Figure 6.

- *l. 72f: As you state, 960mb is an estimate, so the comparison of the pressure minima in simulations with this value should be put in relation and not seen as the absolute truth.*

We agree with the reviewer, and have edited this sentence to say 'is more consistent with', rather than 'better matches'.

- l. 85ff: Please add 1-2 sentences with more information about the added data and especially the improved data assimilation to the main text. It would be good to at least have an idea about the improvements without having to read the Appendix, which should then be for readers with further interest.

Noted. Some additional text on the number of locations added, and the assimilation improvements has been added.

- Figure 3 and elsewhere: Please consider putting the colorbar labels right next to the colorbar.

We have added labels to these colourbars.

- *l. 218: Please elaborate the "simpler grid point approach". Do you mean you simply make the tool independent of neighbouring grid points, hence you could use it on every grid point independently? Please discuss shortcomings of this approach.*
- *l. 221ff: When do you define a member to show precursors: Is this already the case for only one grid point? How do these percentages compare to the probabilities in Fig. 6?*

The text has been edited to make this clearer. We simply count the number of ensemble members with at least one grid point indicating a DSCAPE precursor. The operational warning system has a more complex approach requiring a larger region (multiple grid points) to have a DSCAPE indicator present. There was also a mistake in the text meaning that the % given in the text did not match the figure – this has been fixed.

- *Figure 6: The difference between 20CRv3 and new data seems to be much smaller than new data and new data + improved assimilation. Can you comment on this? Could the improved assimilation be more important for the improvement than the new observations? However, this is not really the case in other figures.*

Unfortunately we do not have an experiment with the change in assimilation scheme applied to the original observations to test some of these issues. We have added: "For the DSCAPE precursor likelihood, there is a clear difference between the experiments that only differ due to the assimilation scheme changes (39% vs 55%). Such differences have been less clear in metrics presented earlier, and we suggest that this may be because DSCAPE is a non-linear threshold-based metric meaning that the reduction in ensemble spread has a larger effect."

- *l. 257: "observed": As in the caption, you should at least mention the HadUK-Grid, i.e., interpolated in-situ observations.*

Agreed – the text has been edited to mention this.

- *Figure 8: Please consider another colour scheme. Furthermore, what is the reasoning behind the 16-84% range?*

We have considered the colour scheme but retained the existing version. We have added the following text to the caption: The 16-84% range is roughly equivalent to showing the ensemble standard deviation but is more appropriate for a non-normally distributed variable such as high-frequency rainfall.

---

## Author Comment (AC3)

**Response to Reviewer 3:**

We thank the reviewer for their careful reading of the manuscript and their thoughtful comments which we discuss below.

- *I am missing more information on the additionally included data (number of new measurements per day etc.), their QC processing. What was the observation error assigned? Also it would be relevant to include marine data in the coverage plots.*

The ship observations available are already included as filled circles in the bottom row of Figure 3, but there are just very few of them! We have added a sentence to the figure caption to clarify this. The cited Craig & Hawkins (2020) paper discusses the collection and QC of most of the new pressure data, but we include more details about the number of stations and their assumed uncertainty in the text and in Appendix A. In the 20CRv3 system the land stations are all assigned an uncertainty of 1.2mb (surface pressure) or 1.6mb (sea level pressure), and ship observations are assigned an uncertainty of 2.0mb. This is unchanged in our experiments.

- *This concerns specifically also the tide gauge record. The authors mention that the QC is not yet done and this work is in process, but it is hard to get a feeling for the error.*

The precision of the tide gauge data is 1 inch and this section of the record has been screened for quality control; we add these details to Appendix A. There remain a few issues with the data from earlier decades which were later to be digitised, which is why the rest of the record has not been published yet, and we can't give a precise answer about the surge statistics. The error on individual data points is hard to constrain without contemporary levelling information, but the coherence of the Liverpool and Hilbre data suggests it is within a few tens of cm.

- *How does the assimilation system digest the quite massive increase in the input? Given the decreased spread, are some of the "original" observations now rejected or vice versa?*

This is an excellent question, and we have gone back to check this. We were already aware of one observation being rejected during the storm in the original 20CRv3 – the existing Stornoway morning observation on 27th February (1002.1mb) appears to be 1 inch/Hg (around 33mb) too high, suggesting a mis-reading or mis-transcription in that record. This is a common feature of such data and that individual observation was not corrected in our experiments. In addition, we discovered that the new data for Nairn was rejected frequently in the reanalysis experiments; this was due to a typo putting Nairn in the incorrect location. This has not negatively affected the reanalysis as the data is rejected, but it is slightly unfortunate as Nairn happens to have been near the centre of Storm Ulysses and the 961mb observation that was taken was rejected due it to being inconsistent with the reanalysis as it was in the wrong place. If it had been in the right place then the uncertainty in the experiments would have been further reduced slightly. The corresponding data file has now been fixed and the filled symbol was not plotted on Figure 3. No other new or existing observations were rejected during the storm and no changes to the manuscript are made.

- *Fig. 3: Is the ensemble mean of the version "with new data" captured in the ensemble spread of the "raw" 20CRv3? This would be interesting for users that*

*cannot rerun 20CRv3. The figures in the Appendix on the spread and RMSE ratio are nice, but do not directly answer the question for this case.*

Thanks. We now highlight in the text that individual members in 20CRv3 are more uncertain in the position of the storm rather than the depth of the storm itself. This leads to the ensemble mean in 20CRv3 showing a storm that is shallower than when the position is constrained by adding additional observations.

- *Say a bit more on the storm surge model already in the main text (at least mention the resolution and start of integration).*

Thanks - we have added some additional text giving more specifics about the storm surge model in the main text and retain the details in Appendix C.

- *L. 367: https://digital.nmla.metoffice.gov.uk/SO_7c59f237-7add-4d78-9c99-4e4210a926e1/ produces a "not found" in my browser*

This link works for us, and points to the Land Observations component of the National Meteorological Archives.

- *The reference list is a bit messy (punctiation, initials, use of "et al.", use of "and", "&" or nothing, etc.).*

Noted. This will be fixed.

---

## Author Response (AR2)

**Technical corrections made to manuscript:**
1) Author affiliations have added countries
2) doi updated for observed data location
3) Additional citation added to introduction about ensemble seasonal forecasts (Walz & Leckebusch 2019)
4) Units changed from mb to hPa, including in figures and in Supp. Info.
5) The Discussion section has been expanded to discuss risk more specifically with added references and an example